# Prognostic Value of the Modified Atherogenic Index of Plasma during Body Mass Reduction in Polish Obese/Overweight People

**DOI:** 10.3390/ijerph16010068

**Published:** 2018-12-27

**Authors:** Jolanta Zalejska-Fiolka, Beáta Hubková, Anna Birková, Beáta Veliká, Beata Puchalska, Sławomir Kasperczyk, Urszula Błaszczyk, Rafał Fiolka, Andrzej Bożek, Beata Maksym, Mária Mareková, Ewa Birkner

**Affiliations:** 1Department of Biochemistry, School of Medicine with the Division of Dentistry in Zabrze Medical University of Silesia, 40-055 Katowice, Poland; bgapuchalska@gmail.com (B.P.); kaslav@mp.pl (S.K.); ublaszczyk@sum.edu.pl (U.B.); rafal.fiolka@med.sum.edu.pl (R.F.); ebirkner@wp.pl (E.B.); 2Department of Clinical and Medical Biochemistry, Pavol Jozef Šafárik University, 040 11 Košice, Slovakia; beata.hubkova@upjs.sk (B.H.); anna.birkova@upjs.sk (A.B.); beata.velika@upjs.sk (B.V.); maria.marekova@upjs.sk (M.M.); 3Clinical Department of Internal Diseases, Dermatology, and Allergology, School of Medicine with the Division of Dentistry in Zabrze Medical University of Silesia, 40-055 Katowice, Poland; andrzej.bozek@sum.edu.pl; 4Department of Pharmacology, School of Medicine with the Division of Dentistry in Zabrze Medical University of Silesia, 40-055 Katowice, Poland; bmaksym@sum.edu.pl

**Keywords:** anthropometric measures, Atherogenic Index of Plasma (AIP), cardiovascular diseases, sub-fractions of HDL-C, LDL-C, weight loss

## Abstract

Although weight loss is recommended for obese patients, it remains questionable how much weight loss is optimal. A novel index that accurately determines the risk of cardiovascular diseases (CVDs) in terms of weight loss is needed. The modified Atherogenic Index of Plasma (AIP), presented here is unique in the literature. It is calculated based on data for anti-atherogenic, high-density lipoprotein cholesterol (HDL-C) fractions, instead of the total HDL-C. This study investigates whether weight loss correlates with CVD risk, and whether the modified AIP allows more accurate diagnostics in obese/overweight people. According to the increase or decrease of AIP during weight loss, 52 Polish patients were subdivided into two groups: group I (increased AIP; *n* = 16) and group II (decreased AIP; *n* = 36). The patients’ body mass composition and fasting serum lipid parameters (total cholesterol, triglycerides, HDL-C, and LDL-C (low-density lipoprotein cholesterol)), and cholesterol in 21 lipoprotein sub-fractions were determined. Over six months, all patients reduced their body mass by about 10%. There were no significant differences in anthropometric measures between groups. Increases in large and intermediate HDL-C fractions 1 to 6 and decreases in smaller fractions 7 to 10 were observed in group II. In group I, HDL-C fractions 1 and 10 decreased, while cholesterol in other fractions increased. Increases were observed in the antiatherogenic HDL-C of 52% of group II and 4% of group I. As for atherogenic HDL-C, a decrease of 24% was observed in group II and an increase of 9% in group I. In group I, increases of very-low-density lipoprotein (VLDL), intermediate-density lipoprotein (IDL), and large LDL fractions were noticed, and the reverse in group II. The results show that the modified AIP is a more accurate indicator of CVD risk than existing indices, and that uncontrolled weight reduction does not necessarily have a beneficial influence, and may adversely affect the cardiovascular system.

## 1. Introduction

Next to genetic predisposition, environmental factors related to improper lifestyles are the main factors leading to disorders of lipids and carbohydrate metabolism, which induce many diseases. Morbidity is lower in people who eliminate or limit a high-calorie, poor quality diet, rich in saturated fatty acids (especially from animal sources), salt, added sugars (sucrose, fructose, or starch), refined grains with a simultaneous increase in the proportion of raw vegetables, fruit, and polyunsaturated fatty acids. Diets rich in fiber have a low glycemic index, and have long been considered as contributing to health promotion. Phenolic compounds represent antioxidants, responsible for prevention of ageing, lipid peroxidation, and DNA damage, among others. Increasing one’s intake of polyunsaturated fatty acids decreases triglycerides. Today’s sedentary lifestyle is characterized by a low level of energy expenditure. Therefore, the consumption of a healthy diet alone does not guarantee good health, but must be balanced by sufficient physical activity [1,2,3]. These modifiable risk factors correlate with the frequency of the occurrence of obesity and of cardiovascular diseases (CVDs) like stroke, heart failure, peripheral arterial disease, or kidney failure [4]. In numerous countries, obesity has grown to an epidemic and become a public health problem. Although weight loss is recommended even in overweight and obese patients with established CVDs, the question of whether such change in body mass is entirely beneficial is controversial [5,6]. Simultaneously, CVDs due to arteriosclerosis, such as stroke and coronary heart disease, are the main causes of morbidity, hospitalization, invalidity, and premature mortality in all high-income and some middle-income countries. For instance, according to World Health Organization (WHO) data, in Poland in 2010 CVDs were the cause of 46.0% of all mortality (51.8% in women and 40.8% in men) and the leading cause of premature mortality (constituting 26.9% of mortality in people before the age of 65 years) [7,8,9,10]. It is estimated that despite significant progress in diagnostics and treatment, in the majority of countries CVDs will remain the main cause of mortality until at least 2030 [11].

Various indices have been used for the diagnosis and prognosis of CVDs. Concentrations of total cholesterol (TC), and its fractions in high-density lipoprotein (HDL) and low-density lipoprotein (LDL) are also frequently cited to justify statin therapy. The Atherogenic Index of Plasma (AIP) is easily calculated from a standard lipid profile. It is a logarithmically transformed ratio of molar concentrations of triglycerides (TG) to high-density lipoprotein cholesterol (HDL-C). The strong correlation of AIP with lipoproteins may explain its high predictive value [12]. It is possible, however, that determination of TC, HDL-C, and low-density lipoprotein cholesterol (LDL-C) concentrations is not sufficient for appropriate medical therapy. LDL-C and HDL-C should be sub-fractionated in order to measure concentrations of large, anti-atherogenic HDL-C (1–3); less atherogenic HDL (4–7); and small, atherogenic particles of HDL-C (8–10), as well as less atherogenic LDL-C (1–2) and atherogenic LDL-C (3–7) lipoproteins. It seems therefore potentially very informative and useful to calculate a modified AIP index, using not the total HDL-C but a concentration of anti-atherogenic subfractions of HDL-C (1–3) in the ratio. Such an approach could be more accurate, not only for diagnosis but for medical therapy as well. We also assumed that body mass reduction is not so important as a decrease of the AIP index for CVD diagnosis. In these ways, the calculation of the new AIP is unique in the literature.

The aim of the study was to investigate whether weight loss correlates with CVD risk, and whether the determination of a new AIP index allows more accurate diagnostics in obese/overweight people.

## 2. Materials and Methods

### 2.1. Patients’ Characteristics

The study was a prospective, one-center study with randomization.

**Study Inclusion Criteria**: body mass index (BMI) > 25 kg/m^2^, with no treatment affecting lipid metabolism; informed consent to participate in the study.

**Study Exclusion Criteria**: lack of consent to participate in the study; severe hepatic, renal insufficiency, respiratory, or circulatory insufficiency; disturbances of consciousness; treatment-resistant depression; chronic alcohol abuse; pregnancy; history of a serious nervous system injury; implanted cardiac pacemaker; treatment affecting lipid metabolism.

Fifty-two patients (16 men and 36 women) from the Metabolic Clinic in Miasteczko Śląskie, Poland qualified for the study from among 300 patients diagnosed as overweight or obese on the basis of medical examination (body mass index (BMI), waist-to-hip ratio (WHR), body fat mass, index of central obesity (ICO)). All patients were characterized by at least one parameter indicating excessive body mass or visceral obesity. None of the patients were taking any medications known to affect lipid metabolism.

### 2.2. Description of Dietary and Physical Activity Recommendations

Patients had started weight reduction therapy based on the reports indicating improving lipid levels after weight loss. All patients before therapy were on a poor-quality diet (high consumption of saturated fatty acids, salt, simple sugars, refined grains, and preservatives) with low intake of antioxidant-rich fruits and vegetables and performed low physical activity. During the first visit, after medical examination and completion of the questionnaire, the individual value of basal metabolism (basic metabolic rate (BMR)), total metabolism (comprehensive metabolic panel (CPM)), and daily energy deficit (DDE) in kcal per day were determined for each patient. We recommended slight calorie restriction (calculated as 15% of energy via CPM). Each patient received personalized, detailed instructions about nutrition and diet composition, according to healthy nutritional recommendations of the Polish National Food and Nutrition Institute (Polish acronym IŻŻ) and World Health Organization (WHO), listed in Table 1. It was also significant to increase vegetable (to 500 g per day) and fruit (to 200/300 g per day) consumption and number of meals to five per day (each of proper composition and caloric value).

In addition to a proper diet, the patients performed moderate physical exercise at least three times a week, lasting 40 min. During follow-up visits (every three weeks), patients were weighed and their waist and hip circumferences were measured. The results of weight loss, as well as the amount and composition of the diet were recorded continuously. Patients were motivated to continue the process, and received support in the form of training in calculating calorie intake and the quality of food products. Weight reduction was monitored until healthy body weight or reduction by 5–15% over initial weight before weight loss was achieved, and lasted six months on average.

### 2.3. Classification of Patients into Study Groups

For some patients, the increase in AIP was observed both before and after therapy. Thus, 52 respondents were categorized into two groups on the basis of change in AIP, calculated according to the regular Formula (1), as
(1)$$\mathrm{AIP}(\mathrm{regular}\;\mathrm{formula})=\log\frac{\mathrm{TG}}{\text{HDC-C}}$$
where TG is the triacylglycerol concentration and HDL-C is the high-density lipoprotein cholesterol.

Subsequently, a modified AIP was calculated, from a new Formula (2)
(2)$$\mathrm{AIP}\left(\mathrm{new}\;\mathrm{formula}\right)=\log\frac{\mathrm{TG}}{\text{anti-atherogenic}\;\text{HDC-C}}$$
where TG is the triacylglycerol concentration and anti-atherogenic HDL-C is the anti-atherogenic high-density lipoprotein cholesterol (subfractions 1–3).

**Group I**: 16 patients (12 Female + 4 Male) with an increase in AIP new formula (in this group no patients had a physiological value of AIP < 0.5 calculated before the therapy).

**Group II**: 36 patients (26 Female + 11 Male) with a decrease in AIP new formula (in this group only one patient had physiological value of AIP < 0.5 calculated before the study).

### 2.4. Biochemical Examination

A biochemical examination was conducted during the first visit and after the therapy blood samples were collected for biochemical tests.

For the biochemical examination, following an overnight fast, blood samples were drawn into tubes and centrifuged the same day to separate serum, which was stored frozen (−80 degrees Celsius) for subsequent analysis. The analysis of the lipoprotein fractions and sub-fractions in the fasting serum with a TC of over 2.59 mmol/L (100 mg/dL) was carried out using the Lipoprint Lipoprotein Sub-fractions Testing System (Quantimetrix, Redondo Beach, CA, United States). The system operates as follows: lipoproteins are stained with a lipophilic dye and separated using high-resolution electrophoretic equipment on a linear polyacrylamide gel. The resolved bands of lipoprotein fractions and sub-fractions are scanned by digital scanner. The generated image is analyzed using Lipoware software, to calculate the cholesterol level for each of the fractions. Liposure control material was used to monitor accuracy and precision. We determined sub-fractions of HDL-C and LDL-C, namely antiatherogenic HDL-C fractions 1–3 (large HDL), less atherogenic HDL-C fractions 4–7 (intermediate HDL), and atherogenic HDL-C fractions 8–10 (small HDL), as well as less atherogenic LDL-C fractions 1–2 (large LDL), atherogenic LDL-C fractions 3–7 (known as cholesterol in small dense LDL (sdLDL)), and cholesterol in sub-fractions of VLDL, IDL-C, IDL-B, and IDL-A.

The parameters of the lipid profile—TC, LDL-C, HDL-C, and TG—were determined using the selective biochemical analyzer BS-200E, (Mindray, Shenzhen, China), and reagents from Alpha Diagnostics (San Antonio, TX, United States). The InBody S10 professional body composition analyzer (InBody, Cerritos, CA, United States) was used to determine body mass composition.

### 2.5. Statistical Analysis

All statistical analyses were performed using SPSS Statistics 22 (IBM, Armonk, NY, United States). Testing for normality was performed by the Kolmogorov Smirnov test. Differences between group means were calculated using a two-sample *t*-test, assuming or not assuming equal variances (based on Levene’s Test for Equality of Variances). The strength of the linear relationship between the two variables was expressed by the Pearson correlation coefficient; a *p* value of <0.05 was assumed to be statistically significant. Means and standard deviations are reported in terms of the original distributions.

The procedures were approved by the Ethics Committee of the Medical University of Silesia, in Katowice, Poland (agreement no. KNW/0022/KB1/19/I/16). Written informed consent was obtained from all the subjects.

## 3. Results

The waist circumference of all qualified persons exceeded the recommended values, except in one patient whose waist circumference was 79 cm, which is borderline. Over all patients, the mean waist circumference was 109.30, however, most of the values exceeded 105 cm; the range was from 79 cm to 135 cm. The value of the WHR was lower than 0.84 in only 13 women and lower than 0.95 in only three men. The average WHR value was 0.90, with a range from 0.72 to 1.09. The BMI ratio in all patients except one exceeded 25. The average was 35.84 and the range was from 24.50 to 52.20. The visceral fat area (VFA) was below 100 cm^2^ in only nine people; the average VFA value was 139.96, and it ranged widely from 63.4 to 210.60. Before the therapy, the mean TG concentration was 132.05 mg/dL, and ranged from 36.00 to 743 mg/dL. Mean total HDL-C concentration was 56.33 mg/dL and also fluctuated within quite a large range, from 31.00 to 92.00 mg/dL.

Because the initial results for each patient were divergent, the results were also presented as the difference between values before body mass reduction (_1) and after body mass reduction (_2); therefore:(3)delta=result after(_2)−result before(_1)
(4)$$\mathrm{percentage}\;\mathrm{change}=\frac{\mathrm{result}\;\mathrm{after}\left(\_2\right)-\mathrm{result}\;\mathrm{before}\left(\_1\right)}{\mathrm{result}\;\mathrm{before}\left(\_2\right)}$$

Anthropometric and biochemical results are found in Table 2, Table 3, Table 4, Table 5 and Table 6. Examples of lipoprotein separation are presented in Figure 1 and Figure 2 for HDL separation and Figure 3 and Figure 4 for LDL separation.

Analyzing the results obtained after dividing patients into two groups by increase or decrease in AIP index, there is no statistically significant difference between study groups for any anthropometric data, whether using the regular or the new AIP formula. There was a non-significantly greater decrease in body mass, waist circumference, and body fat mass (BFM), but not in hip circumference and VFA in group II than in group I. There were no observed differences in calculated anthropometric indices—that is, WHR, BMI, or ICO (Table 3). These indices appear to be unsuitable for our study on the CVD risk of patients. The only parameter that is different for both groups is AIP. Both for the regular AIP formula and the new one, a statistically higher AIP index was observed in group II before the study then for the regular AIP and the new AIP formula, respectively; we observed increases of AIP by 0.05 and 0.09 in group I and decreases by 0.16 and 0.29 in group II (Table 2).

A significant difference in TC levels before, but not after, the therapy was noticed between the two groups. In group II, the concentration of TC was significantly higher (231.58 ± 46.61 mg/dL) in comparison to group I (194.63 ± 40.50 mg/dL). The original AIP TC before in group I was 202.47 ± 44.34, and in group II it was 230.38 ± 56.57; after therapy, TC in group I was 231.35 ± 37.60, and in group II it was 221.85 ± 72.50 (no significant difference). After therapy there was observed a significant increase of TC in group I by about 14% (*p* value of the paired *t*-test = 0.149), and a decrease in group II of about 2% (*p* value of the paired *t*-test = 0.034) (Table 4). There were also differences in the TG concentration between study groups: the concentration of TG in group II was significantly higher before therapy (152.46 ± 116.99 mg/dL) than that in group I (90.19 ± 38.26 mg/dL). The original AIP TG in group I was 89.88 ± 38.93, and in group II it was 151.42 ± 115.22; there was a significant difference between groups, with *p* = 0.006. After therapy, TG was 119.29 ± 44.56 in group I and 105.90 ± 75.43 in group II, with no significant difference. In group II, despite the higher starting value of TG, its concentration decreased about 20%, probably explaining the decreased risk of atherogenicity. In group I, in contrast, an increase of about 27.50% in TG was observed (Table 4); thus, despite the good TG starting poit and body mass reduction, atherogenicity had increased after therapy in group I. Average values in LDL-C before therapy were significantly higher (144.37 ± 33.18 mg/dl) in group II compared to group I, but after therapy there were no significant differences between the groups (original AIP LDL-C before the therapy in group I was 129.06 ± 36.49, and in group II it was 142.00 ± 43.14; after the therapy, in group I, LDL-C was 149.50 ± 31.67, and in group II it was 138.86 ± 62.77, with no significant difference). Despite this, LDL-C decreased in group II by about 0.23% (*p* = 0.059), and increased significantly in group I by about 17%. In both groups, and before and after the therapy, the total HDL-C concentration was similar; before therapy, it reached 56.56 ± 10.42 in group I and 55.32 ± 9.53 in group II, and after therapy, it was 61.97 ± 9.90 and 60.75 ± 11.69 (original AIP HDL-C before the therapy in group I was 55.41 ± 12.36, and in group II it was 56.69 ± 10.11; after therapy in group I, HDL-C was 61.99 ± 13.19, and in group II it was 61.48 ± 11.09, with no significant difference). Although the concentration was higher after the intervention, there were no statistically significant differences between groups (Table 4).

Detailed information was given by the electrophoresis of HDL-C and LDL-C lipoproteins (Table 5 and Table 6).

The results obtained for the regular formula are presented in Table 6. There was observed an increase of small HDL-C in group I of about 1%, while for group II there was observed a decrease of small HDL-C of about 22%. A decrease in HDL10 of about 2% in group II and of about 0.12% in group I was also noticed. In addition, a significantly higher HDL 6 concentration was observed in group II before therapy. We did not get any information about HDL (1–3) using the regular AIP formula. However, there was observed a significant decrease of VLDL in group II (about 9 mg/dL), and an increase of 1.35 mg/dL in group I—the same tendency we observed in IDL-B concentration. It increased about 3.29 mg/dL in group I, and decreased about 1.75% in group II.

The use of the modified new AIP formula (Table 3) allowed us to obtain more information about small and large HDL (Table 5). In particular, the obtained results for group II show that the HDL1 concentration was lower before therapy (2.86 ± 2.40); despite this, there was a significant increase in the HDL1 level after therapy of about 57%. In contrast, in group I there was observed a significant decrease in HDL1 levels after therapy of about 21%. In addition, after therapy a significant increase in HDL2 of about 33% in group II and a decrease of about 3% in group I; as well as an increase of HDL3 of about 3 mg/dL in group II and about 1.56 mg/dL in group I were noticed.

Therefore, after therapy there were observed significant increases in the large, antiatherogenic HDL (1–3) sub-fraction in group II of about 52%, and in group I, increases of about 4%.

We also obtained some significant results for less atherogenic sub-fractions HDL6 and 7. There was observed a significant decrease of HDL7 in group II of about 0.39 mg/dL, and an increase in group I of about 0.31 mg/dL, although the HDL6 and HDL7 concentrations were higher in group II before therapy.

The obtained results show a decrease in HDL8, 9, and 10 sub-fractions in group II, respectively, about 0.64, 0.58, and 2.17 mg/dL. Meanwhile, we observed an increase of HDL8 and 9 in group I, respectively, of about 0.25 and 0.19 mg/dL, with an increase in HDL10 of about 0.13 mg/dL. In summation, we observed an increase of small HDL by about 9% in group I and decrease of small HDL by about 24% in group II.

For less atherogenic sub-fractions of HDL (4–7), no statistically significant differences were found for the regular AIP formula.

Both the regular and the new AIP formula gave us information regarding less atherogenic LDL (1–2). The changes were similar. Using the regular AIP formula, in group I, we observed a significant increase of LDL (1–2) of about 51%, and a decrease of about 6% in group II (Table 6). Using the new AIP formula, a significant increase in LDL (1–2) of about 50% in group I and a decrease of about 5% in group II were noticed. In both cases, LDL-C (1–2) was significantly higher in group II before therapy—the opposite of what we expected. The old AIP formula gives us information about small LDL3 atherogenic particles. In the beginning of the study, the concentration of LDL3 was lower in group I (0.76 ± 2.22 mg/dL) and significantly higher in group II (3.33 ± 6.73 mg/dL). Similar results were obtained after therapy (1.06 in group I and 1.78 in group II), but the delta LDL3 was not significant, probably because of the high standard deviation in group II.

We obtained similar results for atherogenic lipoprotein VLDL, using both old and new AIP formulas. Using the new formula, we observed an increase in VLDL of about 2.56 mg/dL in group I and a decrease in VLDL of about 8.69 mg/dL in group II. These results correlate with atherogenicity.

Using the new AIP formula, significant differences between groups were found in the IDL-B concentration before therapy; it was significantly higher in group II. There were no significant differences after therapy.

## 4. Discussion

One of the most important consequences of visceral obesity (VO) is the development of cardiovascular diseases (CVD), including atherosclerosis and ischemic heart disease. It has been shown that people with excessive visceral fat are exposed to tissue insulin resistance and atherogenic dyslipidemia, with low HDL-C and high concentrations of TG, small dense LDL, and apolipoprotein B [13,14].

The most commonly used determinant of VO is the value of waist circumference. The WHO recommends that waist circumference not exceed 94 cm in men and 80 cm in women. Simultaneously, a waist circumference of over 102 cm in men and over 88 cm in women should be considered as an indication for body mass reduction therapy. However, it has also been noticed that waist circumference, especially in men, correlates in a higher extent with the overall amount of adipose tissue than with the amount of visceral fat [15,16]. In recent years, a new parameter to measure the amount of visceral fat has emerged: the visceral fat area (VFA), obtained by measurement of body composition using body analyzers that employ impedance. The value of VFA over 100 cm^2^ is an indicator of weight reduction. This is a simple and generally available method allowing large-scale research towards counteracting the development of visceral obesity [17,18].

The Framingham Heart Study and Jackson Heart Study, large clinical trials that used computed tomography scan to assess fat distribution, showed that excess visceral fat tissue combined with excess fat located in the liver, heart, and chest positively correlate with the occurrence of metabolic disorders and CVDs. It was also found that CVD is independent of the total amount of fat contained in the body and the amount of subcutaneous fat. In addition, magnetic resonance showed a very close relationship between the amount of fat located in the liver and risk factors for CVDs. High bioavailability of fats increases the production of high-triglyceride-rich VLDL molecules and reduces the metabolism of apolipoprotein B by hepatocytes [13,19]. This was confirmed by Lopez-Jimenez et al., who claimed that among the numerous pathologic mechanisms involved in the initiation and development of coronary atherosclerosis, the most important is the excessive activation of the sympathetic system; the elevated concentrations of atherogenic lipoproteins, TG, and free fatty acids (FFA); increased circulating blood volume; chronic inflammation; and increased tendency to thrombotic changes, which are connected with obesity [20]. The NHANES III study, based on a meta-analysis of prospective clinical observations that included a total of over 220,000 people from 17 countries, indicates that VO determined on the basis of waist circumference measurements is not an independent risk factor for myocardial infarction. However, the incidence of myocardial infarction correlates with the components of the metabolic syndrome, such as hypertension, high TG concentration, and low HDL-C concentration. These findings have shown that none of the parameters used in the diagnosis of obesity—BMI, waist circumference, or WHR ratio—correlates with the prevalence of CVDs if the analysis includes measurements of blood pressure, serum lipid levels, and if carbohydrate metabolism disorders were present [21].

Martínez-González et al. have pointed out that a Mediterranean diet based on vegetables rich in unsaturated fatty acids and polyphenols may be a sustainable and ideal model of CVD prevention [22]. Dietary fat intake has an influence on lipoprotein cholesterol fractions, and hence on atherogenic properties of blood. Based on the results of Hamulka et al. [23], this effect is particularly pronounced in the elderly. The authors recommend measuring both the lipoprotein cholesterol fractions and the AIP.

It is well known that consumption of a healthy diet alone does not guarantee good health, but must be balanced by a sufficient physical activity. Another very important approach is the speed of the weight loss. A gradual, slow reduction of body mass (0.25–0.5 kg a week) is the most effective approach, because it has the important effect of fostering good physical and mental health, and is more likely to lead to maintaining the reduced body weight. Five to ten per cent weight loss in obese people results in an improvement in serum lipid parameters, as well as in glycemia. In addition, a decrease in arterial pressure has also been observed [24,25,26,27,28,29]. Lavie et al. and Nguyen et al. showed that a slow weight reduction achieved by a healthy diet combined with exercise results in a decrease in atherogenic lipids, systolic and diastolic blood pressure, and reduced mortality [30,31]. On the contrary, the rapid weight reduction causes only minor changes in blood pressure.

All patients included in our study were provided a personalized diet and exercise plan. They reduced body mass by an average of 10%, with a simultaneous improvement of anthropometric parameters over a period of six months. AIP, based on the regular formula, did not indicate atherogenicity (AIP over 0.5), nor in one group before weight reduction. Group I consisted of patients with an increase in AIP, while group II consisted of patients with a decrease of this parameter, despite the reduction in weight, BMI, and VFA. After the therapy, despite the increase in atherogenicity based on AIP calculations using the regular formula in 16 patients of group I, the AIP still exhibited physiological values. However, biochemical results, mostly LDL-C and TC, did not indicate that those people were free from CVD risk. This prompted us to accurately determine the sub-fractions of LDL-C and HDL-C with a subsequent adjustment of the regular AIP formula, for a more accurate establishment of the blood atherogenicity.

Applying the new AIP formula, we pointed out that with the exception of one patient, everyone had pathological AIP before the therapy. Whereas AIP in the group of patients (group I) in whom atherogenicity increased during therapy, had initial values close to physiological levels (0.65 ± 0.26), AIP values of the patients in group II were significantly increased (1.02 ± 0.37). The change in atherogenicity to higher values in patients in the first group was negligible (delta 0.09 ± 0.08), and had as a consequence the pathological AIP value. The change of AIP during therapy in patients of group II did not trigger the physiological values, but was significant (delta −0.29 ± 0.18).

Without calculating AIP based on the new formula, our obese patients would be evaluated as CVD risk-free. More detailed knowledge on LDL-C, but especially on HDL-C subfractions enables a more accurate determination of the AIP. Thus, it is the more accurate index for determining CVD risk in overweight and obese people.

This study has several limitations. The small number of patients included in the study is based on the fact, that most patients with obesity are treated for congenital disorders. Our aim was to investigate the effect of weight loss on atherogenesis, without the influence of drugs. There was a difference in the number of patients in the respective study groups. This limitation was due to the strict study inclusion criteria and the classifications based on AIP. We are also aware of the lack of information about the duration of obesity before the weight loss therapy, as most patients in the questionnaire stated that they suffer from obesity long-term, without giving more accurate evidence of developing obesity. We lack information on metabolic disorders, chronic inflammation, and total antioxidant status, even they are known to be independently associated with CVD risk and may play an important role in the heterogeneous effects of obesity on prognosis. After obtaining and analyzing the data regarding these variables. we will be able to draw more precise conclusions.

## 5. Conclusions

The results of our research show that it is certainly necessary to take into account CVD risk before recommending patients for weight reduction therapy. Thus, it is necessary to look for a parameter that accurately determines CVD risk. Because in the new AIP formula we used only anti-atherogenic HDL subfractions (1–3), it is a more accurate indicator of CVD risk than the regular formula. We also found that weight reduction does not necessarily have beneficial effects, and may adversely affect the cardiovascular system. Perhaps weight reduction should not be forced, and better would be to help patients maintain their weight and to convince them to gradually introduce healthy food products (vegetable, fruits, and fish oils) in place of unhealthy ones (animal fats, simple sugar, fast foods, ready meals, and semi-prepared foodstuffs, including frozen meals). The study is in progress.

## Figures and Tables

**Figure 1 ijerph-16-00068-f001:**
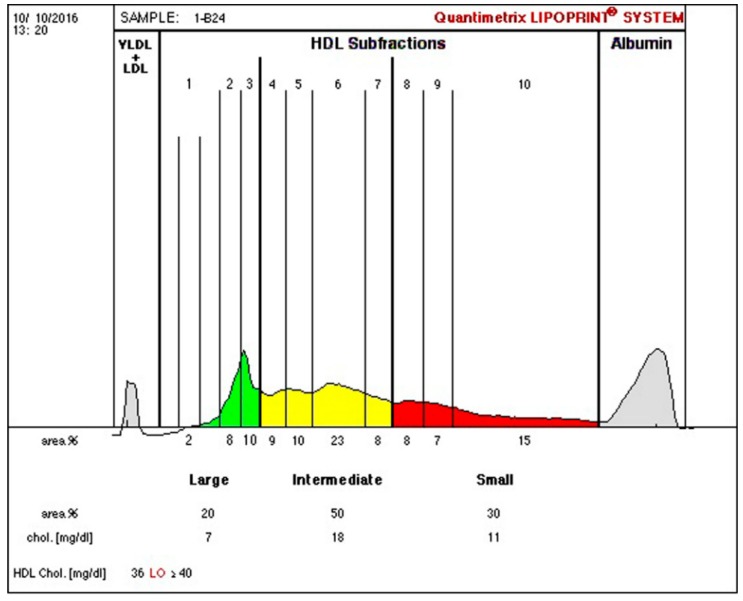
High-density lipoprotein cholesterol (HDL-C) separation before therapy, indicating 20% of large (green), 50% of intermediate (yellow), and 30% of small (red) HDL sub-fractions.

**Figure 2 ijerph-16-00068-f002:**
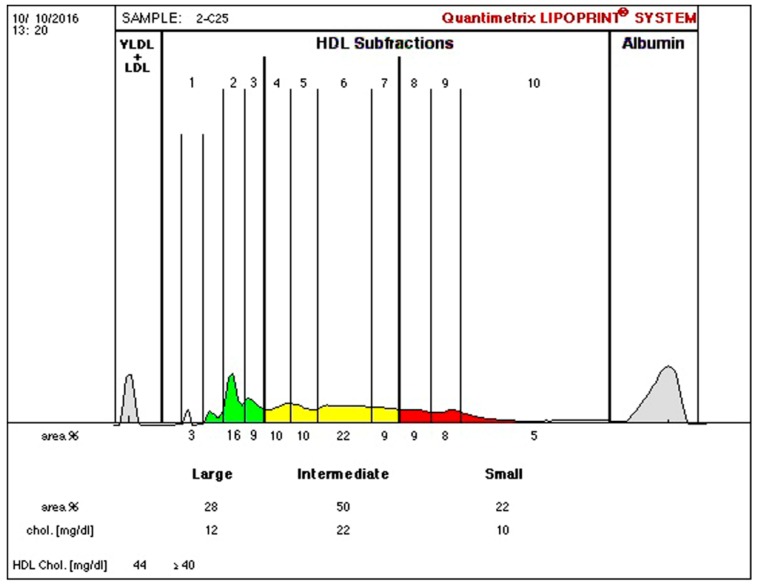
HDL-C separation after therapy, indicating 28% of large (green), 50% of intermediate (yellow), and 22% of small (red) HDL sub-fractions.

**Figure 3 ijerph-16-00068-f003:**
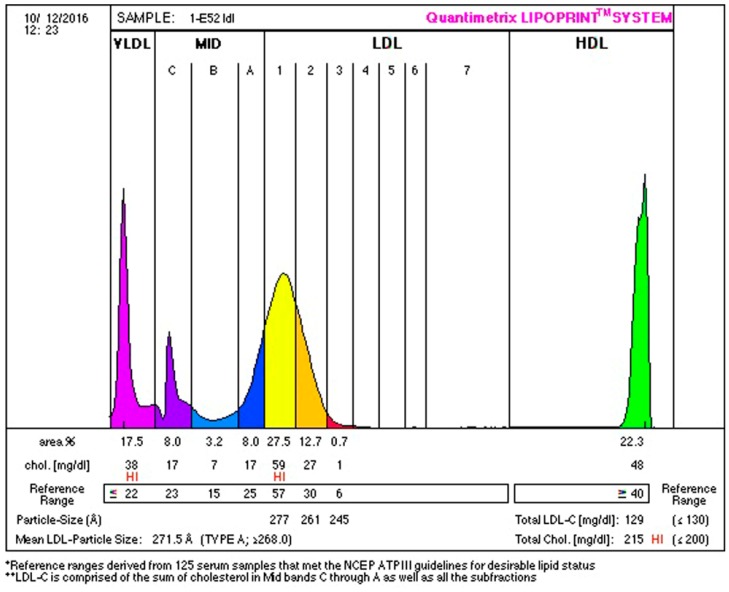
Low-density lipoprotein cholesterol (LDL-C) separation before therapy, indicating 17.5% of very-low-density lipoprotein (VLDL; purple), 19.2% of intermediate-density lipoprotein (IDL; blue), 40.2% of large (yellow and orange), 0.7% of small (red) LDL sub-fractions, and 22.3% of HDL sub-fractions.

**Figure 4 ijerph-16-00068-f004:**
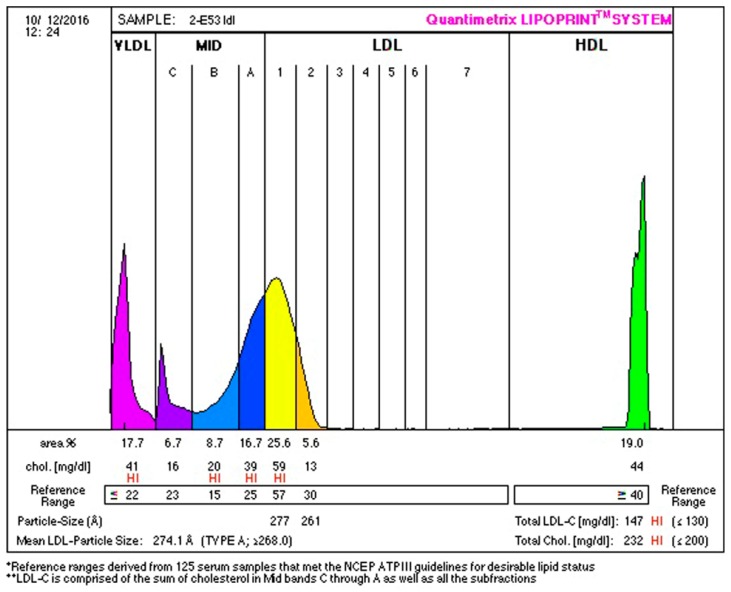
LDL-C separation after therapy, indicating 17.7% of VLDL (purple), 32.1% of IDL (blue), 31.2% of large (yellow and orange), 0.0% of small (red) LDL sub-fractions, and 19.0% of HDL sub-fractions.

**Table 1 ijerph-16-00068-t001:** Diet composition according to healthy nutrition recommendations of the Polish National Food and Nutrition Institute (Polish acronym IŻŻ) and World Health Organization (WHO).

Nutrient	% of Total Energy Intake	Note
fats	25–35%	limiting intake of saturated fatty acids to less than 10% of total energy intake, and intake of 3–6% mono- and polyunsaturated fatty acids in the form of vegetable oils and fish oils
carbohydrates	45–65%	limiting intake of free sugars to less than 10% of total energy intake
proteins	10–15%	animal and vegetable sources

**Table 2 ijerph-16-00068-t002:** Categorization of patients on the basis of change in Atherogenic Index of Plasma (AIP).

	Group I *n* = 16	Group II *n* = 36	*p*
	Mean ± SD	Mean ± SD	
AIP (regular formula)_1	0.18 ± 0.21	0.37 ± 0.27	0.011 *
AIP (regular formula)_2	0.23 ± 0.21	0.22 ± 0.25	ns
delta AIP (regular formula)	0.05 ± 0.10	−0.16 ± 0.14	≤0.001 ***
AIP (new formula)_1	0.65 ± 0.26	1.02 ± 0.37	≤0.001 ***
AIP (new formula)_2	0.75 ± 0.29	0.73 ± 0.30	ns
delta AIP (new formula)	0.09 ± 0.08	−0.29 ± 0.18	≤0.001 ***

Group I: patients with an increase in AIP; Group II: patients with a decrease in AIP; AIP: atherogenic index of plasma; SD: standard deviation; *p*: *p* value of a two-sample *t*-test, assuming or not assuming equal variances based on Levene’s Test for Equality of Variances; *: *p* ≤ 0.050; ***: *p* ≤ 0.001; _1: value before body mass reduction; _2: value after body mass reduction; ns—not significant result.

**Table 3 ijerph-16-00068-t003:** Average values of anthropometric parameters.

	Group I *n* = 16	Group II *n* = 36	*p*
Age (years)	49.06 ± 10.07	47.42 ± 11.05	ns
Height (cm)	166.31 ± 6.78	167.14 ± 9.09	ns
Weight_1 (kg)	95.91 ± 22.11	102.46 ± 20.37	ns
Weight_2 (kg)	87.94 ± 18.79	92.24 ± 19.30	ns
**% change in weight**	**−7.84 ± 4.84**	**−8.89 ± 7.56**	**ns**
Waist circumference_1 (cm)	106.75 ± 18.10	111.03 ± 13.26	ns
Waist circumference_2 (cm)	97.88 ± 14.59	99.78 ± 14.54	ns
**% change in waist circumference**	**−7.88 ± 6.35**	**−10.25 ± 5.80**	**ns**
Hip circumference _1 (cm)	120.06 ± 18.78	122.31 ± 11.77	ns
Hip circumference _2 (cm)	113.31 ± 13.88	113.83 ± 11.40	ns
**% change in hip circumference**	**−5.06 ± 5.38**	**−6.92 ± 5.09**	**ns**
WHR_1	0.89 ± 0.11	0.91 ± 0.08	ns
WHR_2	0.86 ± 0.09	0.88 ± 0.09	ns
**% change in WHR**	**−2.75 ± 5.46**	**−3.69 ± 4.97**	**ns**
ICO_1	0.64 ± 0.10	0.66 ± 0.08	ns
ICO_2	0.59 ± 0.09	0.60 ± 0.08	ns
**% change in ICO**	**−7.88 ± 6.35**	**−10.22 ± 5.82**	**ns**
BMI_1	34.64 ± 7.66	36.57 ± 5.86	ns
BMI_2	31.73 ± 6.68	32.91 ± 5.85	ns
**% change in BMI**	**−8.13 ± 4.53**	**−10.00 ± 5.66**	**ns**
VFA _1	139.53 ± 39.71	141.14 ± 32.25	ns
VFA _2	117.95 ± 31.81	118.79 ± 32.33	ns
**% change in VFA**	**−13.69 ± 15.99**	**−15.26 ± 15.47**	**ns**
BFM (body fat mass)1	38.46 ± 15.28	41.03 ± 14.06	ns
BFM (body fat mass)2	32.21 ± 10.34	34.26 ± 12.36	ns
**% change in BFM**	**−8.36 ± 17.77**	**−16.47 ± 21.61**	**ns**

Group I: patients with increase in Atherogenic Index of Plasma; Group II: patients with decrease in Atherogenic Index of Plasma; *p*: *p* value of a two-sample *t*-test assuming or not assuming equal variances based on Levene’s Test for Equality of Variances; _1: value before body mass reduction; _2: value after body mass reduction; WHR: waist-to-hip ratio; ICO: index of central obesity; BMI: body mass index; VFA: visceral fat area; BFM: body fat mass.

**Table 4 ijerph-16-00068-t004:** Average values of lipid parameters.

	Group I *n* = 16	Group II *n* = 36	*p*
TC_1 (mg/dL)	194.63 ± 40.50	231.58 ± 46.61	0.008 **
TC_2 (mg/dL)	217.58 ± 39.57	226.43 ± 66.26	ns
**% change in TC**	**+13.75 ± 18.73**	**−1.64 ± 19.19**	**0.010 ****
TG_1 (mg/dL)	90.19 ± 38.26	152.46 ± 116.99	0.044 *
TG_2 (mg/dL)	110.63 ± 45.75	111.84 ± 76.16	ns
**% change in TG**	**+27.50 ± 33.66**	**−20.11 ± 28.13**	**≤0.001 *****
HDL-C_1 (mg/dL)	56.56 ± 10.42	55.32 ± 9.53	ns
HDL-C_2 (mg/dL)	61.97 ± 9.90	60.75 ± 11.69	ns
**% change in HDL-C**	**+10.75 ± 11.94**	**+10.50 ± 15.12**	**ns**
LDL-C_1 (mg/dL)	120.06 ± 33.51	144.37 ± 33.18	0.019 *
LDL-C_2 (mg/dL)	137.07 ± 33.97	143.09 ± 58.02	ns
**% change in LDL-C**	**+16.60 ± 32.04**	**−0.23 ± 26.45**	**ns**

Group I: patients with an increase in the Atherogenic Index of Plasma; Group II: patients with decrease in Atherogenic Index of Plasma; *p*: *p* value of a two-sample *t*-test, assuming or not assuming equal variances based on Levene’s Test for Equality of Variances; *: *p* ≤ 0,050; **: *p* ≤ 0,010; ***: *p* ≤ 0,001; _1: value before body mass reduction; _2: value after body mass reduction; TC: total cholesterol; TG: triacylglycerol; HDL-C: high-density lipoprotein-cholesterol; LDL-C: low-density lipoprotein-cholesterol.

**Table 5 ijerph-16-00068-t005:** Average values of statistically significant HDL-C and LDL-C sub-fractions in study groups, categorized according to the new AIP formula.

	Group I *n* = 16	Group II *n* = 36	*p*
HDL1_1	5.13 ± 2.47	2.86 ± 2.40	0.003 **
delta HDL1	−1.50 ± 2.00	0.86 ± 2.60	0.002 **
% change in HDL1	−20.60 ± 41.17	56.90 ± 127.84	0.003 **
HDL2_1	8.81 ± 3.04	6.69 ± 2.99	0.023 *
delta HDL2	0.19 ± 1.97	1.44 ± 2.06	0.045 *
% change in HDL2	3.43 ± 24.80	33.53 ± 46.49	0.019 *
delta HDL3	1.56 ± 1.86	2.92 ± 1.73	0.014 *
delta large HDL (1–3)	0.69 ± 4.66	5.08 ± 4.16	0.001 **
% change in large HDL (1–3)	4.33 ± 26.36	52.32 ± 55.00	≤0.001 ***
HDL6_1	10.50 ± 1.67	12.61 ± 1.84	≤0.001 ***
HDL7_	3.56 ± 0.73	4.22 ± 0.93	0.015 *
delta HDL7	0.31 ± 0.70	−0.39 ± 0.99	0.014 *
delta HDL8	0.25 ± 0.93	−0.64 ± 1.22	0.013 *
delta HDL9	0.19 ± 0.91	−0.58 ± 1.11	0.018 *
delta HDL10	−0.13 ± 1.93	−2.17 ± 3.01	0.016 *
delta small HDL	0.38 ± 3.65	−3.53 ± 4.44	0.003 **
% change in small HDL	8.99± 36.70	−23.68± 31.77	0.002 **
VLDL_1	32.38 ± 15.10	43.19 ± 18.71	0.047 *
delta VLDL	2.56 ± 14.07	−8.69 ± 11.42	0.004 **
IDL-B_1	11.50 ± 5.49	15.39 ± 6.35	0.039 *
delta LDL1	8.81 ± 17.38	−3.03 ± 17.53	0.029 *
LDL2_1	11.94 ± 11.28	21.67 ± 14.20	0.019 *
delta LDL2	4.06 ± 16.27	−5.36 ± 11.76	0.022 *
LDL (1–2)1	53.69 ± 21.60	72.61 ± 25.08	0.012 *
delta LDL (1–2)	12.88 ± 28.79	−8.39 ± 23.77	0.007 **
% change in LDL (1–2)	50.06 ± 89.17	−5.20 ± 28.69	0.027 *

Group I: patients with increase in Atherogenic Index of Plasma—new formula; Group II: patients with decrease in Atherogenic Index of Plasma—new formula; *p*: *p* value of a two-sample *t*-test assuming or not assuming equal variances based on Levene’s Test for Equality of Variances; *: *p* ≤ 0.050; **: *p* ≤ 0.010; ***: *p* ≤ 0.001; _1: value before body mass reduction; _2: value after body mass reduction; HDL (1–10): high-density lipoprotein-cholesterol subfractions (1–10); large HDL: sum of large subfractions of high-density lipoprotein-cholesterol, subfractions (1–3); small HDL: sum of small subfractions of high-density lipoprotein-cholesterol, atherogenic subfractions (8–10); LDL (1–2): less atherogenic low-density lipoprotein-cholesterol subfractions (1–2); VLDL: very low-density lipoprotein; IDL-B: intermediate-density lipoprotein-cholesterol, subfraction B.

**Table 6 ijerph-16-00068-t006:** Average values of statistically significant HDL-C and LDL-C sub-fractions in study groups, categorized according to the AIP regular formula.

	Group I *n* = 16	Group II *n* = 36	*p*
HDL6_1	11.00 ± 1.90	12.53 ± 2.01	0.011 *
delta HDL10_2	0.12 ± 1.80	2.25 ± 3.01	0.009 **
delta small HDL (8–10)	0.18 ± 3.45	3.68 ± 4.89	0.011 *
% change in small HDL (8–10)	−1.19 ± 33.99	22.41 ± 35.77	0.026 *
delta VLDL	−1.35 ± 9.90	8.81 ± 13.86	0.009 **
delta IDL-B	−3.29 ± 8.63	1.75 ± 6.02	0.017 *
delta LDL1	−10.00 ± 12.98	3.89 ± 18.46	0.007 **
LDL2_1	13.12 ± 11.77	21.50 ± 14.18	0.039 *
delta LDL2	−5.94 ± 14.70	5.08 ± 14.38	0.013 *
LDL (1–2)_1	57.41 ± 24.75	73.14 ± 26.74	0.046 *
delta LDL (1–2)	−15.94 ± 22.33	8.97 ± 26.26	0.001 **
% change in LDL (1–2)	−51.43 ± 81.46	6.34 ± 32.79	0.011 *
LDL3_1	0.76 ± 2.22	3.33 ± 6.73	0.045 *

Group I: patients with increase in Atherogenic Index of Plasma—new formula; Group II: patients with decrease in Atherogenic Index of Plasma—new formula; *p*: *p* value of a two-sample *t*-test assuming or not assuming equal variances based on Levene’s Test for Equality of Variances; *: *p* ≤ 0,050; **: *p* ≤ 0,010; ***: *p* ≤ 0,001; _1: value before body mass reduction; _2: value after body mass reduction; HDL (1–10): high-density lipoprotein-cholesterol subfractions (1–10); large HDL: sum of large subfractions of high-density lipoprotein-cholesterol, subfractions (1–3); small HDL: sum of small subfractions of high-density lipoprotein-cholesterol, atherogenic subfractions (8–10); LDL (1–2): less atherogenic low-density lipoprotein-cholesterol subfractions (1–2); VLDL: very low-density lipoprotein; IDL-B: intermediate-density lipoprotein-cholesterol, subfraction B.

## Data Availability

The database of aggregated statistics prepared for analysis is stored in secure, confidential, password protected storage in the server of the Medical University of Silesia. The data has been anonymized. Completely de-identified records could be made available to interested persons/organizations on request to the corresponding author at jzalejskafiolka@sum.edu.pl.

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
