# Peer review of "Prognostic Value of the Modified Atherogenic Index of Plasma during Body Mass Reduction in Polish Obese/Overweight People"

_ijerph, 2018, doi:10.3390/ijerph16010068_

Round 1

Reviewer 1 Report

The manuscript ID: ijerph-406361 submitted by Zalejska-Fiolka et al. is interesting and addressed an important point for public health. The manuscript is well presented and the experimental design correct. Importantly, it also contains a section with the limitations of the study.

I have some comments to improve the manuscript value.

1) The tittle could be more powerful. If authors truly think that their new formula to calculate AIP is a better way to calculate CVD risk, that information should be highlighted in the title.

2) Authors could also consider to include in the tittle (and abstract) that the study was done with Polish patients.

3) The AIP regular and new formulas should be shown in a clear way. So, the reader can easy identify the differences at the first sight.

4) Quality (resolution) of figures 1-4 is low and need to be improved.

5) Table organization and appearance could be improved. Also tables 2-4 lack legends.

6) Method section should contain subheadings and no data.

7) The discussion is long and the author contribution section should be filled.

Author Response

Response to Reviewer 1 Comments

Dear Reviewer,

We would like to thank you very much for your careful and constructive review. We have corrected the manuscript according to your recommendations and suggestions. We hope that you will find the changes we have made as appropriate.

Point 1. The tittle could be more powerful. If authors truly think that their new formula to calculate AIP is a better way to calculate CVD risk, that information should be highlighted in the title.

Response 1: Thank you very much for suggestion. We truly think that this formula is a better way to use AIP for diagnosis but we decided not to change the title, since the determination of anti-atherogenic fractions of HDL-C is still not a standardized method.

Point 2. Authors could also consider to include in the tittle (and abstract) that the study was done with Polish patients.

Response 2: We have added this information in the title and abstract; “Prognostic value of modified Atherogenic Index of Plasma during body mass reduction in Polish obese/overweight people”

Point 3. The AIP regular and new formulas should be shown in a clear way. So, the reader can easy identify the differences at the first sight.

Response 3: We have inserted formulas for the calculation of AIP according to the regular formula as well as from a new formula.

Point 4. Quality (resolution) of figures 1-4 is low and need to be improved.

Response 4: We improved the resolution of the figures.

Point 5. Table organization and appearance could be improved. Also tables 2-4 lack legends.

Response 5: We have changed the formatting of the tables and we've put a legend under each table.

Point 6. Method section should contain subheadings and no data.

Response 6: In “Materials and Methods” section we added subheadings, as Patients’ characteristic; Description of dietary and physical activity recommendations; Classification of patients into study groups, Biochemical examination; Statistical analysis.

Point 7. The discussion is long and the author contribution section should be filled.

Response 7: We have corrected the discussion and Author contribution was filled.

Reviewer 2 Report

The manuscript presented by Zalejska-Fiolka et al., is an interesting study. The general manuscript addresses aspects of interest in public health and clinical nutrition. Delivering valuable information regarding the population (Polish Population?).

The manuscript has an adequate structure and order. The experimental design is adequate, the results are interesting and support the discussion. However, before accepting the manuscript it is necessary that the authors make the following modifications.

Major comments

1. The introduction should include a better description of what fatty acids and carbohydrates increase or reduce the risk of developing cardiovascular disease or other chronic diseases.

Lines 44. Lipids (saturated fatty acids) and carbohydrates (sucrose, fructose or starch)

Suggested references

Pearson-Stuttard et al. Comparing the effectiveness of mass media campaigns with price reductions targeting fruit and vegetable intake on US cardiovascular disease mortality and race disparities. Am J Clin Nutr. 2017; 106 (1): 199-206.

Anand et al., Food Consumption and its Impact on Cardiovascular Disease: Importance of Solutions Focused on the Globalized Food System: A Report From the Workshop Convened by the World Heart Federation. J Am Coll Cardiol. 2015; 66 (14): 1590-1614.

2. The Introduction is several lines is very reiterative, the authors should improve this section. It is necessary to include aspects related to diet, especially the gain of body weight in the introduction.

Suggested reference

Mozaffarian et al., Changes in diet and lifestyle and long-term weight gain in women and men. N Engl J Med. 2011; 364 (25): 2392-404.

3. In material and methods, the authors should include as a specific sub-section the "statistical analysis".

4. Regarding the statistical analysis, the authors evaluated the normal distribution of the data ?. Include this point in that sub-section.

5. Why the difference in the number of subjects per experimental group (group I = 16 v / s group II 36). The authors should indicate this aspect as a limitation of the study.

6. The quality figures (1 - 4) and resolution of the images is not good. It is necessary to improve the figures.

7. Tables 2, 3 and 4 do not have legends. This is a major mistake. Please correct

8. On lines 383 and 385, "<0.5" is observed, is this correct, or should it be "<0.05"?

9. Question. The authors have data on food intake, energy, and macronutrients of the subjects?

10. Considering the contribution of the study and to improve the quality of the manuscript it is necessary that the authors improve the writing of the discussion. Especially in the aspects related to the gain of corporal weight, alimentary interventions, and protection of the health. In addition to discussing how certain food groups or food components might have influenced the results of the study

Suggested references

Martínez-González et al., Benefits of the Mediterranean Diet: Insights From the PREDIMED Study. Prog Cardiovasc Dis. 2015; 58 (1): 50-60.

Hernandez-Rodas et al., Relevant Aspects of Nutritional and Dietary Interventions in Non-Alcoholic Fatty Liver Disease. Int J Mol Sci. 2015; 16 (10): 25168-98.

Minor comments

1. The title should indicate that the study was in the Polish population.

2. In the tables replace "St. Dev "by S.D.

3. Better the format of the tables. All must follow the same format.

Author Response

Response to Reviewer 2 Comments

Dear Reviewer,

We would like to thank you very much for your careful and constructive review. We have corrected the manuscript according to your recommendations and suggestions. We hope that you will find the changes we have made as appropriate.

Point 1. The introduction should include a better description of what fatty acids and carbohydrates increase or reduce the risk of developing cardiovascular disease or other chronic diseases.

Lines 44. Lipids (saturated fatty acids) and carbohydrates (sucrose, fructose or starch)

Suggested references

Pearson-Stuttard et al. Comparing the effectiveness of mass media campaigns with price reductions targeting fruit and vegetable intake on US cardiovascular disease mortality and race disparities. Am J Clin Nutr. 2017; 106 (1): 199-206.

Anand et al., Food Consumption and its Impact on Cardiovascular Disease: Importance of Solutions Focused on the Globalized Food System: A Report From the Workshop Convened by the World Heart Federation. J Am Coll Cardiol. 2015; 66 (14): 1590-1614.

Point 2. The Introduction is several lines is very reiterative, the authors should improve this section. It is necessary to include aspects related to diet, especially the gain of body weight in the introduction.

Suggested reference

Mozaffarian et al., Changes in diet and lifestyle and long-term weight gain in women and men. N Engl J Med. 2011; 364 (25): 2392-404.

Response 1 and 2: We have analyzed the suggested references and changed the introduction adding relevant information on the impact of nutrition on the development of CVDs and obesity.

Point 3.  In material and methods, the authors should include as a specific sub-section the "statistical analysis".

Response 3: In “Materials and Methods” section we added subheadings, as Patients’ characteristic; Description of dietary and physical activity recommendations; Classification of patients into study groups, Biochemical examination; Statistical analysis.

Point 4. Regarding the statistical analysis, the authors evaluated the normal distribution of the data? Include this point in that sub-section.

Response 4: All statistical analyses were performed using SPSS Statistics 22 (IBM, Armonk, New York, USA). Testing for Normality was performed by the Kolmogorov Smirnov test. Subsequently, a p value of a two-sample t-test assuming or not assuming equal variances was established based on Levene’s Test for Equality of Variances. The information was added in the sub-section Statistical analysis

Point 5. Why the difference in the number of subjects per experimental group (group I = 16 v/s group II 36). The authors should indicate this aspect as a limitation of the study.

Response 5: Among the 300 patients with a diagnosis of overweight or obesity met the inclusion criteria of the study 52 patients. On the basis of change in AIP value, the patients were classified into two sub-groups. The distribution was not uniform in the study groups.

Point 6. The quality figures (1 - 4) and resolution of the images is not good. It is necessary to improve the figures.

Response 6: We have improved the resolution of the figures.

Point 7. Tables 2, 3 and 4 do not have legends. This is a major mistake. Please correct

Response 7: We have changed the formatting of the tables and we've put a legend under the tables.

Point 8. On lines 383 and 385, "<0.5" is observed, is this correct, or should it be "<0.05"?

Response 8: Values “˂ 0.5 and ˃ 0.5” are the AIP reference values.

Point 9. Question. The authors have data on food intake, energy, and macronutrients of the subjects?

Response 9: Yes, we have such information. Food intake (energy, fats, carbohydrates, vegetables, fruits, macronutrients) was monitored during each follow-up visits (every 3 weeks).

For each patient, the value of basal metabolism (basic metabolic rate, BMR), total metabolism (comprehensive metabolic panel, CPM) and daily energy deficit (DDE) in kcal per day were determined. We recommended slight calorie restriction (calculated as CPM – 15% of energy).

Each patient received personalized detailed instructions about nutrition and a composed diet, according to healthy nutrition recommendations of the Polish National Food and Nutrition Institute (Polish acronym IŻŻ) and World Health Organization (WHO). According to the recommendations diet consisted of 25%–35% fats (limiting intake of saturated fatty acids to less than 10% of total energy intake, and intake of 3%–6% mono- and polyunsaturated fatty acids in the form of vegetable oils and fish oils), 45%–65% of carbohydrates (limiting intake of free sugars to less than 10% of total energy intake), and 10%–15% proteins (animal and vegetable sources). It was also significant that we recommended increase of the number of meals to four or five per day, each of proper caloric value.

The information is added to the sub-section Description of dietary and physical activity recommendations

Point 10. Considering the contribution of the study and to improve the quality of the manuscript it is necessary that the authors improve the writing of the discussion. Especially in the aspects related to the gain of corporal weight, alimentary interventions, and protection of the health. In addition to discussing how certain food groups or food components might have influenced the results of the study

Suggested references

Martínez-González MA, Salas-Salvadó J, Estruch R, Corella D, Fitó M, Ros E. Benefits of the Mediterranean Diet: Insights From the PREDIMED Study. Prog Cardiovasc Dis. 2015; 58 (1): 50-60.

Hernandez-Rodas MC, Valenzuela R, Videla LA.Relevant Aspects of Nutritional and Dietary Interventions in Non-Alcoholic Fatty Liver Disease. Int J Mol Sci. 2015; 16 (10): 25168-98.

Response 10: We have changed the discussion according suggested aspect.

Minor comments

Point 1. The title should indicate that the study was in the Polish population.

Response 1: We have changed the title

Point 2. In the tables replace "St. Dev "by S.D.

Response 2: We replaced it.

Point 3. Better the format of the tables. All must follow the same format.

Response 3: We have changed the formatting of the tables.
